# Hospital Pharmacy Professionals and Cardiovascular Care: A Cross-Sectional Study Assessing Knowledge, Attitudes, and Practices in Saudi Arabia

**DOI:** 10.3390/healthcare12060630

**Published:** 2024-03-11

**Authors:** Fahad Alzahrani, Reem A. Alhusayni, Nosaiba B. Khairi, Ammar A. Bahauddin, Shadi Tamur

**Affiliations:** 1Department of Pharmacy Practice, College of Pharmacy, Taibah University, Madinah 42353, Saudi Arabia; 2Scientific Research Unit, College of Pharmacy, Taibah University, Madinah 42353, Saudi Arabia; reem_alhussini@taibahu.edu.sa (R.A.A.); nosaibakhairy@taibahu.edu.sa (N.B.K.); 3Department of Pharmacology and Toxicology, College of Pharmacy, Taibah University, Madinah 42353, Saudi Arabia; abahauddin@taibahu.edu.sa; 4Department of Pediatric, College of Medicine, Taif University, Taif 21944, Saudi Arabia; shaditamur@tu.edu.sa

**Keywords:** knowledge, attitude, practice, barriers, cardiovascular disease, hospital pharmacist, prevention, management, barriers

## Abstract

Background: In Saudi Arabia, cardiovascular diseases (CVDs) establish a significant health challenge, with a high prevalence and substantial impact on mortality and disability burden. Evaluating the knowledge, attitude, and practices (KAPs) of hospital pharmacy professionals towards CVDs prevention and management is crucial for effective healthcare strategies. Methods: A cross-sectional, multicenter study was conducted using a self-administered survey targeting hospital pharmacists and pharmacy technicians in the Madinah region of Saudi Arabia. The survey assessed their KAPs towards CVDs prevention and management, incorporating demographic variables and perceived barriers. Results: Out of 177 contacted pharmacy professionals, 159 (89.8%) completed the survey. The study results revealed inadequate knowledge levels with an average score of 3.87 out of 7, indicating significant gaps in comprehending drug interactions, managing lipid levels, and addressing resistant hypertension. Attitudes were generally positive towards CVDs prevention. Practices in CVDs prevention were satisfactory but varied, with notable gaps in providing educational materials and collaboration with other healthcare professionals. Major barriers included time constraints, lack of private counseling areas, and low patient expectations regarding pharmacy professionals’ roles. Conclusion: The study has uncovered notable deficiencies in cardiovascular care, especially within the realm of hospital pharmacy professionals in Saudi Arabia’s Madinah region. This finding underscores the importance of implementing specialized educational initiatives and ongoing professional development programs for these healthcare workers. By focusing on these areas and overcoming the challenges identified, we can significantly improve the contribution of hospital pharmacy professionals in Saudi Arabia toward the prevention and management of cardiovascular diseases.

## 1. Introduction

Cardiovascular disease (CVD), which is commonly known as heart disease, continues to pose a major threat to global health, encompassing a range of conditions which may affect the heart and circulatory system [1]. The World Health Organization reports that CVD accounts for approximately one-third of global mortality, with an estimated 17.9 million deaths annually [2]. The prevalence of CVD has been steadily increasing, with the number of cases nearly doubling from 271 million in 1990 to 523 million in 2019 [3]. Furthermore, the number of CVD-related deaths has shown an upward trend, rising from 12.1 million in 1990 to 18.6 million in 2019 [3]. 

In Saudi Arabia, the prevalence of CVD presents a significant health challenge. In 2016, there were approximately 201,300 Saudi nationals living with CVD, with 149,600 adults diagnosed with ischemic heart disease and 51,700 individuals suffering from cerebrovascular disease. The impact of CVD is further highlighted by the fact that it accounts for over 45% of all deaths in the country [4,5]. 

The established link between reducing cardiovascular disease (CVD) risk factors and improved outcomes in CVD management is well documented [6]. Guidelines advocate for a proactive approach to mitigate these risks and prevent major cardiovascular events. Pharmacy professionals play a crucial role in preventing cardiovascular diseases, both primary and secondary. In addition to their traditional responsibilities of dispensing medications, pharmacy professionals are now more involved in direct patient care activities, including educating patients on medication use and managing diseases, which complement the efforts of physicians. These interventions are pivotal in promoting medication adherence, achieving therapeutic goals, ensuring the safe use of medications, and enhancing overall patient care [7,8]. 

In modern healthcare, hospital pharmacy professionals are often responsible for more complex tasks, such as implementing clinical guidelines, adjusting dosages, and titrating medication. On the other hand, community pharmacy professionals tend to focus on improving patient education and adherence, which may also involve monitoring patient outcomes. This division of labor highlights the complementary roles each pharmacy professional plays in patient care. Hospital pharmacy professionals concentrate on clinical management, while community pharmacy professionals focus on patient engagement and education [8].

A recent investigation has verified that community pharmacy professionals in Saudi Arabia have a deep understanding and awareness of the risk factors associated with cardiovascular disease (CVD) [9]. Additionally, these professionals demonstrate a proactive approach to CVD risk evaluation, underscoring their willingness to play a significant role in the management of cardiovascular health [10]. However, our review of the existing literature indicates a gap in the assessment of knowledge, attitudes, and practices regarding the primary prevention of CVD among hospital pharmacy professionals. Our study aims to address this gap by delving into the knowledge, attitudes, and practices of hospital pharmacy professionals in the Madinah region of Saudi Arabia, with a focus on the holistic prevention and management of cardiovascular diseases.

## 2. Materials and Methods

### 2.1. Study Design and Eligibility Criteria

A multicenter cross-sectional study was conducted for three months starting from June 2023 in hospitals located in the Madinah region of Saudi Arabia. A self-administered survey was distributed targeting hospital pharmacy professionals to assess their knowledge, attitudes, and practices toward CVD prevention and management.

The study includes licensed pharmacists and pharmacy technicians from the Saudi Commission for Health Specialties who work full-time in hospitals inside Madinah region. Pharmacy professionals who declined to take part in the study, who did not complete all the survey questions, or who were unavailable at the hospital site throughout the data collection period were excluded from the study. 

The survey data were gathered using an electronic questionnaire hosted on Google Forms. Among the 246 hospital pharmacy professionals registered across the ten major hospitals in the Madinah region, 177 consented to participate in the survey, with 159 completing it in its entirety (response rate was 89.9%). 

### 2.2. Survey Instrument and Distribution

Pharmacy professionals were recruited by convenience sampling. Recruitment and consent were obtained in person, while the survey itself was self-administered online. Data collection was carried out by six trained research assistants to minimize investigator bias. The research assistants received training on standardized data collection protocols and survey administration procedures prior to the initiation of the study. To ensure consistency, all research assistants followed the same data collection techniques and survey administration procedures.

A link to the questionnaire was given to pharmacy professionals who agreed to participate in the study and were asked to complete it while the investigators were on duty to explain and clarify any details about the questionnaire. For pharmacy professionals who did not have time for the first visit, the investigators asked the participants if they preferred to schedule another appointment at their convenience so that the participants could continue the study at a later date, or if they could be contacted via phone or text messages (WhatsApp) to complete the questionnaire, inquire about questions related to the questionnaire questions, and solve any problem they face.

The survey questionnaire tool was meticulously developed through an iterative process, drawing upon the existing literature [10,11,12,13,14,15], the clinical expertise of our research team, and discussions with collaborating CVD researchers from various institutions across Saudi Arabia. 

The final structure of the questionnaire contains five sections, starting with the sociodemographic section, which consists of twelve items. The second section contains seven items to test basic knowledge about CVD. In the third section, on current practice in cardiovascular prevention or management, eight items are included in the set, and the four Likert scales range from “never” to “always”. The fourth section is about attitudes toward cardiovascular health promotion and the management of cardiovascular risk factors, which includes ten items with a similar scale to the previous section of four responses ranging from “strongly agree” to “strongly disagree”. The fifth section, focused on pharmacists’ perceived barriers to providing cardiovascular disease prevention services, contains eleven items with five Likert scales ranging from “not considerable” to “highly considerable”.

To evaluate the cardiovascular knowledge of the participants, each correct response was awarded 1 point, while incorrect responses received 0 points. The highest achievable knowledge score was 7. Pharmacy professionals’ overall knowledge scores were classified according to Bloom’s 1968 cut-off criteria [16], with scores of 5.6–7 deemed good, a score of 4.2–5.5 being moderate, and scores below 4.2 considered poor. Practice scores were derived from responses coded from “never” to “always” on a scale of 0 to 3, with a top score of 24 signifying high practice. Attitude scores were determined by responses ranging from “strongly disagree” to “strongly agree”, scored from 1 to 4, with the highest possible score of 40 reflecting a positive attitude (Table 1).

### 2.3. Validity and Reliability of the Study Questionnaire

To improve content validity, the survey tool was reviewed by a panel of four experts in survey design and cardiovascular pharmacotherapy. The panel included two clinical pharmacists, one academic pharmacist, and one cardiologist. Each item of the questionnaire was assessed for its relevance using a Likert scale ranging from one to five, where a rating of one indicated unimportance and a rating of five denoted high importance. Items that were deemed irrelevant by all four researchers were excluded from the questionnaire, while those considered relevant or highly relevant were retained. In cases where there were any ambiguous items, the researchers engaged in thorough discussions and reached a consensus through agreement. This meticulous review process ensured that the questionnaire included only the most pertinent and meaningful items, enhancing the validity and reliability of the study’s findings [17].

Prior to conducting the official survey, a group of nine pharmacists was randomly selected to test the questionnaire in terms of clarity and comprehension. It is important to note that these pharmacists were not part of the final study. The feedback received from this pilot testing indicated that the questionnaire was precise and did not require any modifications. The test–retest method was employed to ensure the stability of the scores obtained from the questionnaire. Twelve pharmacists and pharmacy technicians were asked to complete the questionnaire twice, with a short interval of 30 min to 1 h between the rounds. This approach aligns with previous studies that have established acceptable coefficients at >80% [18,19]. Additionally, Cronbach’s alpha was utilized to test the internal consistency of the questionnaire items. To ensure internal consistency, Cronbach’s alpha should be above 80% [20]. The internal consistency of the items used in the test was good, as indicated by Cronbach’s Alpha of 85.4%. The questionnaire that was utilized has been tested and confirmed to be reliable and validated.

### 2.4. Statistical Analysis

The statistical analysis was carried out using S.P.S.S. version 28 (I.B.M. Co., Armonk, NY, USA). Numerical data were presented in two ways: either as the median and interquartile range (IQR) for skewed data or as the mean and standard deviation (S.D.) for normally distributed variables. Categorical data were presented as the frequency and percentage. Multiple linear regression analysis was performed to evaluate different factors associated with knowledge score. The assumptions necessary for multiple linear regression, including homoscedasticity, the absence of multicollinearity, no significant outliers, normal distribution, and independence of residuals, have all been met. A two-tailed *p* value of less than 0.05 was considered statistically significant.

### 2.5. Ethical Approval and Consent to Participate

The study proposal has been reviewed and approved by the Institutional Review Board (I.R.B.) General Directorate of Health Affairs in Madinah, with the identification number (H-03-M-84). The informed consent form was given to pharmacy professionals at the beginning of the questionnaire.

## 3. Results

### 3.1. Demographic and Response Rate

The findings of the survey showed that the majority of pharmacy professionals are aged between 31 and 41 years (50.3%). The gender distribution is relatively balanced, with a slight male predominance (52.8% male, 47.2% female). The majority of pharmacists have work experience of 2–5 years (34.6%), and the most common degree held by them is a Bachelor of Pharmacy (72.3%). Tertiary hospitals employ the most significant number of respondents (45.3%). In terms of practice setting, the most common setting is inpatient care (54.1%). The majority of respondents work more than 8 h per day (66.0%). A significant majority of respondents (74.8%) process more than 15 prescriptions per workday. The median number of cardiovascular disease patients seen per workday is 10, with a wide range of 0–200. The majority of respondents encounter CVD patients in the range of 25–50% (42.1%). Approximately 47.2% of respondents reported previous training in CVD prevention and management. The detailed sociodemographic characteristics of the hospital pharmacy professionals are shown in Table 2.

### 3.2. Pharmacy Professionals’ Self-Reported Knowledge about CVD

The study assessed cardiovascular knowledge among respondents using seven-item questions with scores ranging from 0 to 7 points. Overall scores indicated a relatively low level of knowledge, with a mean of 3.87 out of 7 (SD = 1.61) correct answers. The percentage of respondents answering each item correctly ranged widely from 44% to 67.9%, indicating significant gaps in knowledge across many of the assessed topics. Specifically, only 44% correctly identified the increased risk of liver dysfunction from combining statins and fibrates, representing an essential gap as this combination is contraindicated explicitly in the guidelines. Knowledge regarding optimal triglyceride levels was also low, with only 58.8% selecting the guideline-recommended target of <150 mg/dL.

Additionally, only 50.7% knew that fibrates provide the superior lowering of triglycerides compared to other agents. Knowledge about the most effective statins for raising HDL was similarly deficient, with only 57.2% selecting rosuvastatin, compared to about 25% choosing each of the other options. Furthermore, only 50.9% correctly identified the diagnostic criteria for resistant hypertension. In contrast, the most significant proportion of participants (67.9%) correctly identified the adverse effects of the ACE inhibitors and ARBs (Table 3).

### 3.3. Pharmacy Professionals’ Practice of CVD Prevention and Management 

The data presented in Table 4 provide insights into the current practices of the participants in CVDs prevention and management. The overall mean total practice score was 11.68 ± 5.11, suggesting that the practice of pharmacy professionals toward CVD prevention and management is generally low. It is noteworthy that a significant proportion of pharmacy professionals have limited involvement in providing educational materials about CVDs prevention and management, with 35.8% rarely offering such materials to patients. However, it is encouraging to note that 40.9% of respondents often respond to patient inquiries regarding CVDs. Furthermore, a considerable number of respondents (67.3%) rarely or never invite other healthcare professionals, such as nurses, dieticians, or physicians, to screen patients for CVD risk factors in the pharmacy setting.

Similarly, 37.7% of respondents stated that they rarely meet with other healthcare professionals to provide advice or counseling on the importance of adopting healthy lifestyles to prevent CVDs. Screening patients for CVD risk factors is often or rarely performed by 35.2% and 34% of respondents, respectively. On a positive note, a considerable proportion of respondents (38.4%, 37.1%, and 35.2%) often serve patients with advice or counseling on adopting healthy lifestyles, the importance of screening and early detection of CVD risk factors, and assess individual CVD risk using cardiac risk assessment tools, respectively. 

### 3.4. Pharmacy Professionals’ Attitudes towards CVD Prevention and Management

The findings presented in Table 5 shed light on the attitudes of pharmacists toward CV prevention and management. The mean total attitude score of 28.21 ± 6.04 indicates generally neutral attitudes toward CVD prevention and management among the respondents. A significant proportion of respondents (50.3%) agreed that pharmacists should be involved in CVD-related health promotion activities in the pharmacy. Integrating CVD-related health promotion into the daily practice of pharmacists was considered necessary by the majority of respondents (57.9%). Additionally, the respondents expressed confidence and preparedness in providing CVD health promotion, with 51.6% agreeing with this statement. Additionally, a considerable proportion of respondents agreed that providing counseling to CVD patients was their responsibility as pharmacists (51.6%). The importance of distributing CVD education materials in the hospital pharmacy was acknowledged by 48.4% of respondents. Lastly, a substantial proportion of respondents agreed that providing CVD counseling to patients can improve their professional state and increase professional satisfaction (49.1%). 

### 3.5. Pharmacy Professionals’ Perceived Barriers to CVD Prevention and Management

The data presented in Figure 1 reveal the most frequently reported barriers to the involvement of respondents in the prevention and management of CVD. Among the identified barriers, lack of time or increased workload was reported by the highest percentage of participants (30.2%). The second most frequently reported barrier was the lack of a private counseling area, with 28.9% of respondents identifying this as a challenge. Furthermore, a considerable proportion of participants (22.6%) reported low patient expectations regarding the role of pharmacy professionals in CVD prevention. 

### 3.6. Multiple Regression for the Factors Associated with Pharmacy Professionals’ Knowledge 

The results of the multiple regression analysis, as presented in Table 6, reveal several significant factors associated with the knowledge score of the respondents (3.87 ± 1.61). These findings provide valuable insights into the factors that influence the knowledge levels of pharmacy professionals in the context of CVD. Firstly, it was observed that pharmacists with a Bachelor of Pharmacy degree had higher knowledge scores compared to pharmacy technicians. The coefficient for this association was 1.01, with a 95% confidence interval ranging from 0.31 to 1.72 and a significant *p*-value of 0.005. Furthermore, clinical pharmacists were found to have a higher knowledge score compared to pharmacy technicians. The coefficient for this association was 2.10, with a 95% confidence interval ranging from 1.11 to 3.08 and a highly significant *p*-value of less than 0.001. Additionally, participants working in tertiary hospitals were found to have a higher knowledge score compared to those working in primary hospitals. The coefficient for this association was 0.71, with a 95% confidence interval ranging from 0.07 to 1.35 and a significant *p*-value of 0.031. Lastly, pharmacists who treat a higher percentage of CVD patients, ranging from 76 to 100% of their total patients, demonstrated a significantly higher knowledge score compared to those who treat a lower percentage of CVD patients, ranging from 0 to 24%. The coefficient for this association was 1.02, with a 95% confidence interval ranging from 0.01 to 2.04 and a significant *p*-value of 0.048. 

## 4. Discussion

This study aimed to examine the knowledge, attitudes, and practices, as well as associated barriers, of hospital pharmacy professionals in the prevention and management of cardiovascular diseases in the Madinah region of Saudi Arabia. The results of the study indicate that pharmacy professionals possess a poor level of overall scores in terms of cardiovascular knowledge, with an average of only 3.87 out of 7 correct answers. This suggests that there are notable gaps in their understanding of essential areas such as drug interactions, lipid management, and resistant hypertension. These findings highlight the need to address the significant knowledge deficiencies in cardiovascular care among pharmacy professionals. Interestingly, these results align with the findings of a study conducted by Al-Ashwal et al., which reported insufficient knowledge regarding CVD management among their study pharmacists [10]. 

Moreover, there is a marked difference in knowledge scores among hospital pharmacy professionals, influenced by their educational attainment. Notably, pharmacy technicians are seen to have low scores in their knowledge assessments, in contrast to pharmacists. This observation aligns with the research conducted by Hu et al., who identified that pharmacists with lower academic achievements, minimal professional status, and lack of training possess a comparatively minimal essential awareness of adverse drug reaction (ADR) reporting [21]. This suggests that pharmacists with higher levels of education and training tend to have a better understanding of managing CVD.

The study indicated that pharmacy professionals’ involvement in CVD prevention and management is generally insufficient, yet there are positive elements, such as patient engagement and lifestyle counseling. Emphasizing the need for enhanced collaboration with healthcare professionals and further training, the study suggests these measures could bolster their contribution to CVD care [10,11,13].

Despite the potential benefits of interprofessional collaboration for improving CVD care quality [22,23,24], the study found that, significantly, the majority of pharmacy professionals seldom involve other healthcare professionals in CVD risk screening due to various barriers, highlighting the need for systemic changes to promote multidisciplinary approaches [12,25].

The study shows that pharmacy professionals have positive attitudes towards CVD prevention and management, with a mean attitude score of 28.21 ± 6.04. Over half of the participants (50.3%) feel confident in their CVD care roles. This eagerness to adopt advanced responsibilities is consistent with trends in Saudi Arabia, where pharmacists are increasingly seeking patient-centered roles [26]. Similar attitudes are noted in other regions, like Qatar and Yemen, where pharmacists acknowledge their crucial role in CVD management. These results emphasize the importance of encouraging such attitudes [10,14]. 

Despite these favorable perceptions towards CVD prevention and management, Saudi pharmacy professionals perceived many barriers commonly encountered in engaging participants in the prevention and management of CVD. These included lack of time, lack of private counseling, and low patient expectations regarding the role of pharmacists in CVD prevention and management. These barriers have already been identified in previous studies [10,14,27,28].

Time constraints or increased workload were the main challenges, cited by 30.2% of participants, indicating that these issues may limit pharmacy professionals’ involvement in CVD prevention and management. To address this, El Hajji et al. suggested hiring more pharmacy technicians and defining their roles clearly [29]. Additionally, 28.9% reported the absence of a private counseling area as a barrier, highlighting the need for confidential spaces for effective patient care [30].

Furthermore, 22.6% noted low patient expectations of pharmacists’ roles in CVD prevention, echoing studies by Charles et al. and findings from Australia on public awareness. These points stress the need for boosting awareness and education on pharmacists’ crucial contributions to CVD care [31,32]. 

### Strengths and Limitations

This survey is a pioneering effort in Saudi Arabia to assess hospital pharmacy professionals’ knowledge, attitudes, practices, and challenges in CVD care. However, it has limitations, including convenience sampling and sole focus on the Madinah region, which may affect generalizability. The use of self-administered surveys risks bias, and its cross-sectional nature limits causal inference. Although we validated our survey through expert review and pilot testing, we needed to employ more comprehensive validation techniques, and voluntary participation may have introduced bias. Future research should aim for broader sampling and include longitudinal studies to establish causality. 

## 5. Conclusions

This study sheds light on hospital pharmacy professionals’ knowledge, attitudes, practices, and challenges in CVD prevention and management in Saudi Arabia’s Madinah region, revealing significant knowledge gaps. Professionals with higher education and training showed better knowledge. The research also highlights pharmacy professionals’ active role in patient consultations and their generally positive view towards CVD care. However, overcoming challenges like time constraints, lack of private consultation areas, and underappreciation of pharmacy professionals’ roles is critical. Addressing these issues through targeted strategies could lead to public health improvements in CVD care, relying on interprofessional collaboration, robust support systems, and patient education to enhance outcomes and strengthen pharmacy professional–patient relationships.

## Figures and Tables

**Figure 1 healthcare-12-00630-f001:**
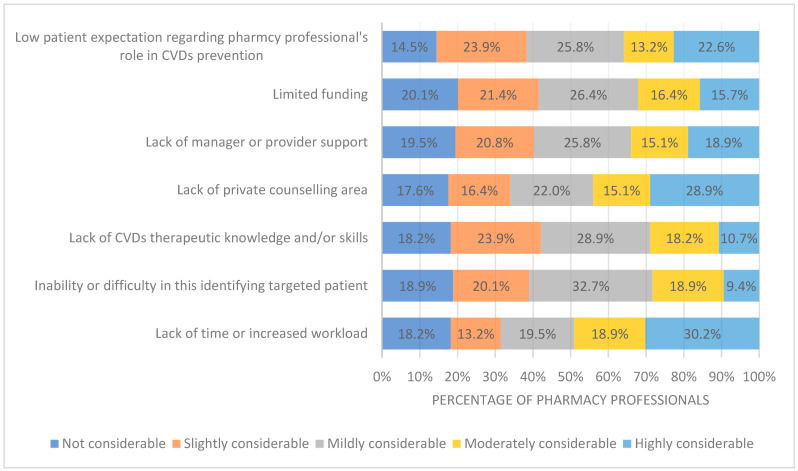
Barriers of pharmacy professionals to involvement in prevention and management of CVDs.

**Table 1 healthcare-12-00630-t001:** Bloom’s cutoff categories for the total knowledge, practice, and attitudes score.

	Category	Sores (%)
Knowledge	good	5.6–7 (80−100%)
	moderate	4.2–5.5 (60−79%)
	poor	<4.2 (<60%)
Practice	high	19.2–24 (80−100%)
	moderate	14.4–19.1 (60−79%)
	low	<14.1(<60%)
Attitude	positive	32–40 (80−100%)
	neutral	24–31 (60−79%)
	negative	<24 (<60%)

**Table 2 healthcare-12-00630-t002:** Sociodemographic characteristics of pharmacy professionals (*n* = 159).

Variable	Category	*n*	%
Age (years)	18–24	6	3.8
	25–30	65	40.9
	31–41	80	50.3
	42–55	5	3.1
	56–65	3	1.9
Gender	Male	84	52.8
	Female	75	47.2
Work experience (years)	≤1	18	11.3
	2–5	55	34.6
	6–10	49	30.8
	>10	37	23.3
Job or role title	Pharmacy technician	20	12.6
	Bachelor of Pharmacy/Doctor of Pharmacy	115	72.3
	Clinical pharmacist	24	15.1
Type of hospital	Primary	34	21.4
	Secondary	53	33.3
	Tertiary	72	45.3
Pharmacy practice setting	Inpatient	86	54.1
	Outpatient	64	40.3
	ER	8	5.0
	Narcotic	3	2.4
	Chemotherapy	5	3.1
	IV/TPN	14	8.8
	Clinical	18	11.3
Duration of service provided per day (hr)	<8	54	34.0
	>8	105	66.0
Prescriptions processed on average per workday	3–5	8	5.0
	6–8	12	7.5
	9–15	20	12.6
	>15	119	74.8
n of people with CVD seen per workday			
Median (IQR)		10 (3–20)	
Range		0–200	
Percentage of CVD patients	0–24	59	37.1
	25–50	67	42.1
	51–75	23	14.5
	76–100	10	6.3
Previous training on CVDs prevention and management (yes)		75	47.2

IV: intravenous; TPN: total parenteral nutrition.

**Table 3 healthcare-12-00630-t003:** Pharmacy professionals’ knowledge of CVDs prevention and management.

Variable	Category	*n*	%
Target blood pressure in patients with diabetes	<120/80 mmHg	33	20.8
	**<130/80 mmHg**	**93**	**58.5**
	<140/90 mmHg	32	20.1
	<150/90 mmHg	1	0.6
The most common adverse effect caused by the antihypertensive drugs ACEIs and ARBs	Hypokalemia	35	22.0
	**Hyperkalemia**	**108**	**67.9**
	Hyperglycemia	9	5.7
	Hyponatremia	6	3.8
	Dry cough	1	0.6
Normal level of triglycerides	<100 mg/dL	41	25.8
	**<150 mg/dL**	**93**	**58.5**
	<200 mg/dL	20	12.6
	<500 mg/dL	5	3.1
Anti-dyslipidemic agents that cause more decreases in triglycerides levels than others	Bile acid sequestrants	12	7.5
	Statins	52	32.7
	**Fibrates**	**80**	**50.3**
	Ezetimibe	15	9.4
The most effective statins for increasing HDL	**Rosuvastatin**	**91**	**57.2**
	Atorvastatin	40	25.2
	Lovastatin	13	8.2
	Simvastatin	15	9.4
Resistance HTN confirms a diagnosis	**Office BP 130/80 mmHg or > and pt taking at least three medications at optimal doses (Confirm Adherence) or office BP of <130/80 mmHg but pt required at least four antihypertensive medications**	**81**	**50.9**
	Office BP 130/80 mmHg or greater and pt taking at least three medications at optimal doses (Regardless Adherence) or office BP of <130/80 mmHg but pt required at least 4 antihypertensive medications	34	21.4
	Office BP 130/80 mmHg or greater and pt taking at least two medications at optimal doses (Confirm Adherence) or office BP of <130/80 mmHg, but pt required at least 4 antihypertensive medications	27	17.0
	Office BP 130/80 mmHg or greater and pt taking at least two medications at optimal doses (Confirm Adherence) or office BP of <130/80 mmHg, but they required at least 3 antihypertensive medications	17	10.7
Combined statin with fibrates is discouraged because of the higher risk of	Heart failure	43	27.0
	**Liver dysfunction**	**70**	**44.0**
	Acute kidney injury	30	18.9
	Hyperglycemia	16	10.1
Total score (Mean ± SD)	3.87 ± 1.61		

Correct answers are presented in **bold**. ACEIs: angiotensin-converting enzyme inhibitors; ARBs: angiotensin receptor blockers; HDL: high-density lipoprotein.

**Table 4 healthcare-12-00630-t004:** Current practice of pharmacy professionals in CVDs prevention or management *(n* = 159).

Variable	Never	Rarely	Often	Always
Provide patients with educational materials about CVDs prevention (educational materials may include brochures, flyers, pamphlets, posters, buttons, and others	40 (25.2%)	57 (35.8%)	48 (30.2%)	14 (8.8%)
Respond to patient inquiries relating to CVDs	11 (6.9%)	33 (20.8%)	65 (40.9%)	50 (31.4%)
Invite other healthcare professionals (e.g., nurse, dietician, physician) to screen patients for CVD risk factors in the pharmacy	53 (33.3%)	54 (34%)	36 (22.6%)	16 (10.1%)
Meet other healthcare professionals (e.g., nurse, dietician, physician) to provide patients with advice or counseling regarding the importance of adopting and maintaining healthy lifestyles to prevent CVDs	27 (17%)	60 (37.7%)	45 (28.3%)	27 (17%)
Screen patients for the presence of CVD risk factors	31 (19.5%)	54 (34%)	56 (35.2%)	18 (11.3%)
Serve patients with advice or counseling regarding the importance of adopting and maintaining healthy lifestyles to prevent CVDs	17 (10.7%)	46 (28.9%)	61 (38.4%)	35 (22%)
Provide patients with advice or counseling on the importance of screening and early detection of CVD risk factors	30 (18.9%)	45 (28.3%)	59 (37.1%)	25 (15.7%)
Assess patients risk for CVDs using cardiac risk assessment tools	36 (22.6%)	50 (31.4%)	56 (35.2%)	17 (10.7%)
Total score (Mean ± SD)	11.68 ± 5.11			

**Table 5 healthcare-12-00630-t005:** The attitude of pharmacy professionals towards CVDs prevention or management (*n* = 159).

Variable	Strongly Disagree	Disagree	Agree	Strongly Agree
Screening for the presence of CVD risk factors is the best use of pharmacy professional time	29 (18.2%)	34 (21.4%)	87 (54.7%)	9 (5.7%)
I should be involved in CVD-related health promotion activities in the pharmacy	14 (8.8%)	35 (22%)	80 (50.3%)	30 (18.9%)
Integrating CVD-related health promotion into my daily practice as a pharmacist is important	9 (5.7%)	27 (17%)	92 (57.9%)	31 (19.5%)
I feel confident and prepared to provide CVD health promotion to my patients	14 (8.8%)	32 (20.1%)	82 (51.6%)	31 (19.5%)
Providing counseling to cardiovascular patients (hypertensive, diabetes) is my responsibility as a pharmacy professional	11 (6.9%)	27 (17%)	82 (51.6%)	39 (24.5%)
Distributing CVD education materials is important in the hospital pharmacy	14 (8.8%)	29 (18.2%)	77 (48.4%)	39 (24.5%)
There is strong evidence to suggest that pharmacy professionals can influence patients to adopt CVD prevention practices	10 (6.3%)	25 (15.7%)	90 (56.6%)	34 (21.4%)
Providing CVD counselling to my patients can improve my professional state and increase my professional satisfaction	11 (6.9%)	27 (17%)	78 (49.1%)	43 (27%)
Patients want me, as a pharmacy professional, to counsel them on CVD prevention	17 (10.7%)	43 (27%)	78 (49.1%)	21 (13.2%)
Patients appreciate my effort as a pharmacy professional to counsel them about CVDs	10 (6.3%)	33 (20.8%)	87 (54.7%)	29 (18.2%)
Total score (Mean ± SD)		28.21 ± 6.04		

**Table 6 healthcare-12-00630-t006:** Multiple linear regression analysis for factors associated with knowledge score.

Variable	Category	Coefficient	95%CI	*p*-Value
Age (years)	18–24	Ref		
	25–30	0.31	−1.20 to 1.81	0.688
	31–41	0.67	−0.96 to 2.29	0.418
	42–55	−0.81	−2.86 to 1.25	0.44
	56–65	0.70	−1.70 to 3.10	0.566
Gender	Male	Ref		
	Female	−0.08	−0.64 to 0.47	0.761
Work experience	≤1	Ref		
	2–5	0.21	−0.75 to 1.18	0.659
	6–10	−0.55	−1.61 to 0.51	0.309
	>10	−0.51	−1.69 to 0.67	0.393
Job or role title	Pharmacy technician	Ref		
	Bachelor of Pharmacy	1.01	0.31 to 1.72	0.005 *
	Clinical pharmacist	2.10	1.11 to 3.08	<0.001 *
Type of hospital	Primary	Ref		
	Secondary	−0.20	−0.88 to 0.48	0.564
	Tertiary	0.71	0.07 to 1.35	0.031 *
Duration of service provide per day (h)	<8	Ref		
	>8	−0.31	−0.86 to 0.25	0.275
Prescriptions processed on average per workday	3–5	Ref		
	6–8	0.09	−1.29 to 1.47	0.900
	9–15	0.23	−1.02 to 1.49	0.712
	>15	0.98	−0.16 to 2.11	0.091
Percentage of CVD patients	0–24	Ref		
	25–50	0.36	−0.18 to 0.91	0.191
	51–75	0.33	−0.42 to 1.08	0.387
	76–100	1.02	0.01 to 2.04	0.048 *
Previous training in CVDs prevention and management		−0.06	−0.56 to 0.44	0.820

CI: confidence interval; *: statistically significant *p* value < 0.05.

## Data Availability

All necessary data are available from the manuscript. The authors will share the available dataset if required.

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
