# Peer review of "Hospital Pharmacy Professionals and Cardiovascular Care: A Cross-Sectional Study Assessing Knowledge, Attitudes, and Practices in Saudi Arabia"

_healthcare, 2024, doi:10.3390/healthcare12060630_

Round 1

Reviewer 1 Report

Comments and Suggestions for Authors

I thank the editor and authors for reviewing the manuscript “Enhancing Cardiovascular Disease Care through Hospital Pharmacists: A Cross-Sectional Study Assessing Knowledge, Attitude, and Practice in Saudi Arabia”.

One general comment relates to the limited capacity of hospital pharmacists, who mainly contact inpatients, to address behavioural risk factors for CVD with the public, compared to community-based professionals. Why did the study not involve community pharmacists from Madinah if CVD prevention was one central theme, assuming a health system organization with high street licensed pharmacies?

More confusing was in line 125, where the authors mentioned primary health centres. Do these have a hospital pharmacist, or was the survey to be answered by pharmacy technicians, as mentioned in line 129? I am unaware of education and training differences and professional responsibilities, but I would guess these differ between professions. If all are ‘pharmacists’, this needs to be explained further, including a change in the manuscript title. Lines 138-139 also require more explanation on how the health system works: do primary hospitals equal healthcare centres, serving outpatients, i.e. ambulatorial patients? Additionally, narcotic and chemotherapy pharmacists were included, although I am not sure how much CVD prevention and management these professionals are obliged to do.

Specific comments:

Lines 133-134. I do not understand the passage “… and pharmacists we were not able to communicate with personally via phone or text message (WhatsApp) were excluded from the study.” Why, on a self-administered survey, do you need direct communication with participants?

Lines 142-143. It seems the recruitment was done in person, although it is not completely clear to me. The authors mention “potential pharmacists”; what do you mean by potential? Were you not sure if they were actual pharmacists? Moreover, which electronic support was used to run the survey?

Lines 178-188. A reference to the validation procedure is missing here.

Lines 189-201. I appreciate the effort to have a valid and reliable survey tool, including face validity and reliability. However, presenting results from a questionnaire with only internal consistency testing is susceptible to criticism. It seems 12 participants were used to estimate Cronbach’s alpha, which looks relatively poor. In addition, if the data is not objective (e.g. numerical) and based on Likert scales, it is hard to guarantee construct validity without further statistics. Some researchers recommend having 5-10 participants per item as a rough guideline.

Line 207. Multilinear regression requires several analytical decisions and tests, for instance, checking assumptions such as linearity and residuals’ homoscedasticity. Nothing is presented that assures the reader that the model and its estimates are trustworthy.

Line 216. I do not understand how “medical care will not change” in this study context.

Lines 267-269. “… considerable number of pharmacists (67.3%) rarely or never invite other healthcare professionals, such as nurses, dieticians, or physicians, to screen patients for CVDs risk factors in the pharmacy setting.” Why should they? Are the others needed? If so, how dependent are pharmacists to achieve patient CVD education and counselling?

Table 3. I would expect to have a maximum score value. There is a line for the total, but I could not find a value.

Lines 279-300. As with previous subheadings, I would advise the authors to avoid extensive repetition of findings already presented in Tables. Maybe authors can highlight from the findings those more relevant to practice.

Line 316. The authors mention a ‘knowledge score’ here. This should be the 3.87 (± 1.61) presented in Table 2. However, it is unclear how the variable was calculated; please describe it in the methods. This applies to the other dependent variables, while variable names should be used systematically throughout the paper.

Lines 447 onwards. While the Discussion is well put, this section would benefit from streamlining. Some concepts are repeated, and no speculative hypotheses are presented. For example, the lack and need to raise awareness is somewhat repeated, with the second mention not bringing a new angle or perspective (lines 445 and 464).

Line 484 and below. This subheading systematically reviews all the limitations presented in the study, particularly those associated with the methods used. While extensive, some limitations are described as if the associated result was expected, e.g. sample power and representativeness, which were never assumed.

Thank you.

Reviewer 2 Report

Comments and Suggestions for Authors

Thank you for your submission.

Unfortunately, I do not believe it has sufficient scientific rigour or novelty for international publication.  

The manuscript is far too long, especially for a very small study (should be about half the current length). The writing style should be much more scientific and concise.

The abstract (conclusions section) should be very clear that the results only apply to a sample of Saudi hospital pharmacists. As is, the statements seem to be referring to all pharmacists worldwide.

The authors repeatedly confuse hospital and community pharmacists throughout the manuscript e.g. they state “Hospital pharmacists, as critical members of healthcare professionals (HCPs), play multifaceted roles in raising awareness among the public” and refer to papers only involving community pharmacists. This occurs several times through the manuscript. Community pharmacists are much more likely to be involved in raising public awareness etc. than hospital pharmacists. Not surprisingly, the results subsequently confirmed the limited involvement of the hospital pharmacists in patient education on CVD prevention and screening etc. These things would, in most parts of the world, be more likely to occur in community pharmacy.

The study includes pharmacy technicians and pharmacists. It is not appropriate to combine the results and call all the respondents “pharmacists”. It is also inconsistent with the article title.

Is work experience in Tabel 1 in years?

What does “clinical pharmacist” mean as a degree in Table 1?

The data have been over-analysed, given the small sample (and need to firstly exclude technicians) and very small numbers in some categories. The authors have simply analysed everything possible, which indicates a poor scientific approach. As a result, there are too many tables and figures. The statistical approach is also incorrect. The use of multiple regression is inappropriate with so low numbers, a high number of variables (9) and some very small cell sizes in Table 1.  It is clear that minimum cell size requirements would not have been met for a valid multiple regression analysis. Furthermore, it is unclear how and whether the issue of multicollinearity between variables was addressed.

Comments on the Quality of English Language

Mostly fine.

Reviewer 3 Report

Comments and Suggestions for Authors

Interesting topic with some important aspects for pharmacy practice.

The paper continues to focus on pharmacists as one category, but the sample included pharmacy technicians.  The paper's discussion, results, and conclusions continue to use the term pharmacist.  This is misleading because pharmacy technicians are included so to be accurate, the data should either only include pharmacists' results or separate the data between pharmacists and technicians.  The title indicates pharmacists as well as the abstract and this is misleading.

Watch the continued use of also and that in the paper.  Reword when possible. 

You report 177 contacted out of total list of ???.  Better explanation of total sample size.  How many did you contact who did not want to participate or who did not respond.  Your convenience sample was 177 with 159 responding.  12.6% were technicians.  This % is high enough to change results for conclusions about pharmacists.  

Limitations are well done and discuss most major problems with the study design.  

What was the difference between mentioned data collectors versus investigators.  Could the investigators introduced bias when collecting data?  How were data collectors trained?  Did everyone do the same thing in collecting information (or answer their questions as they filled out the Google Form).  

Be very careful in making statements about enhancing validity and reliability.  Reliability was established with Cronbach's alpha, but validity is not really discussed fully.  Researchers clearly had face validity with experts, but what other validity measures were done?  Better description.  

Stats analysis.  Some concerns with categorical data such as a Likert scale.  Authors indicated data are either mean or median if skewed.  What data were skewed and where is the median.  The best representation of Likert data would be median with IQR.  Likert data is not normally distributed and should not be presented as such.  

Table 2.  Please use ADA terminology.  Target blood pressure in patients with diabetes (not diabetic)

Better explanation of overall total scores for the tables and results.  

Not sure how to interpret a mean of 3.87 for knowledge out of 7?  Is this good or bad or average?  Would I expect a technician to have the same knowledge as a clinical pharmacist--no.  So why is the average score combined?

Table 3, mean total score, not representative --how to interpret?  Same with Table 4

Section 3.5  mean score??  13.95?  interpretation?

With multiple regression analysis, were you able to provide a predictive equation for your outcomes rather than just correlation coefficients.  

Page 15 lines 447 to 455  Did your study prove this?  Did you prove remarkable levels of motivation.  You looked at attitude, knowledge, and practice--how does this correlate to motivation and willingness?  Are the pharmacists well-prepared to address specific CVD needs--you had many areas of knowledge gaps?? Are patients more likely to receive?? not sure you proved this with the study?

Good limitations and conclusion is good, but needs a little tighter control to what was actually studied. Some of those things listed should be more in the area of future studies and future directions.  

Comments on the Quality of English Language

Minor edits to improve

Round 2

Reviewer 2 Report

Comments and Suggestions for Authors

Still too long and lacking any novelty.  The reassurances given around the validity of the statistical analyses are not convincing and backed by any evidence.

Comments on the Quality of English Language

Mostly Ok.

Reviewer 3 Report

Comments and Suggestions for Authors

Table #4 format is not in a readable form with the PDF so I was unable to evaluate the revised table.  Figure 1 still says % of Pharmacists rather than change to Pharmacy Professionals.  

Page 15 lines 425-440 remove firstly, secondly, thirdly, fourthly, and lastly. Line 448 remove On a brighter note (not technical language).

Paper is much improved, more accurate conclusions and limitations

Comments on the Quality of English Language

comments made in authors section
